# Frame Structure Fault Diagnosis Based on a High-Precision Convolution Neural Network

**DOI:** 10.3390/s22239427

**Published:** 2022-12-02

**Authors:** Yingfang Xue, Chaozhi Cai, Yaolei Chi

**Affiliations:** School of Mechanical and Equipment Engineering, Hebei University of Engineering, Handan 056038, China

**Keywords:** structural health monitoring, fault diagnosis, vibration signal, convolution neural network, anti-noise capability

## Abstract

Structural health monitoring and fault diagnosis are important scientific issues in mechanical engineering, civil engineering, and other disciplines. The basic premise of structural health work is to be able to accurately diagnose the fault in the structure. Therefore, the accurate fault diagnosis of structure can not only ensure the safe operation of mechanical equipment and the safe use of civil construction, but also ensure the safety of people’s lives and property. In order to improve the accuracy fault diagnosis of frame structure under noise conditions, the existing Convolutional Neural Network with Training Interference (TICNN) model is improved, and a new convolutional neural network model with strong noise resistance is proposed. In order to verify THE superiority of the proposed improved TICNN in anti-noise, comparative experiments are carried out by using TICNN, One Dimensional Convolution Neural Network (1DCNN) and First Layer Wide Convolution Kernel Deep Convolution Neural Network (WDCNN). The experimental results show that the improved TICNN has the best anti-noise ability. Based on the improved TICNN, the fault diagnosis experiment of a four-story steel structure model is carried out. The experimental results show that the improved TICNN can obtain high diagnostic accuracy under strong noise conditions, which verifies the advantages of the improved TICNN.

## 1. Introduction

Frame structure includes various kinds of supports, skeletons, and shaped frames, which are widely used in mechanical engineering, civil engineering, aerospace, medicine, and other fields [1,2,3,4]. In the process of use, frame structure collapses due to bolt loosening, uneven stress, material fatigue, and oxidation [5], which leads to major engineering accidents and immeasurable losses to people’s lives and property. Installing vibration sensors on frame structure to collect real-time running state information of frame structure and using the information to carry out effective health monitoring and fault diagnosis of frame structure can not only ensure the safety of frame structure itself, but also ensure the normal operation of machinery, building stability, aviation safety, and emergency treatment; therefore, it is of great scientific and practical significance to propose an appropriate fault diagnosis method to carry out accurate fault diagnosis and predict the healthy running state of frame structure in advance.

Fault diagnosis is a process to discover and identify the fault type of equipment or systems. The purposes of fault diagnosis include fault detection, fault type judgment, fault location, etc. Traditional fault diagnosis methods include Fourier transform [6], wavelet transform [7], empirical method [8], etc. Although traditional fault diagnosis methods have been widely used in different fields, the performance of fault diagnosis declines when facing the object with complex fault content, large data volume, and strong noise interference. Facing the shortcomings of traditional fault diagnosis methods, many scholars have introduced BP neural network and SVM into fault diagnosis to improve accuracy of fault diagnosis. You et al. used BP neural network to diagnose faults of cutting tools; however, because the fitting ability of BP neural network depends on the stability of input and output, it is difficult to use for the fault diagnosis of frame structure [9]. Compared with BP neural network, SVM has higher classification accuracy. For this reason, Gao et al. used SVM to study the fault diagnosis of transformer and obtained higher fault diagnosis accuracy [10]. However, SVM requires high signal purity, so it is difficult to carry out an accurate fault diagnosis for frame structure under a strong noise environment. It can be seen from the above that the traditional methods can effectively diagnose the fault of simple frame structure when the frame structure is complex with strong noise and multiple diagnostic targets, the above methods are not only complex in operation, but also difficult to achieve ideal accuracy.

With the secondary development of artificial intelligence, deep learning theory has been widely used in natural language processing [11,12,13], text information retrieval [14,15,16], image processing [17,18,19], computer vision [20,21,22], and other fields, so various convolutional neural networks are derived from these applications. Among them, One Dimensional Convolution Neural Network (1DCNN) [23], VGG-net [24], Rest-net [25], and Inception [26] have been used by many researchers for fault diagnosis [27] and health monitoring in various fields by transforming dimensions, changing depth, changing the size of convolution kernel, etc. In the aspect of fault diagnosis, intelligent fault diagnosis of rotating machinery based on deep learning is a research hotspot and has been widely studied and applied. Zhang et al. obtained a neural network named First Layer Wide Convolution Kernel Deep Convolution Neural Network (WDCNN) [28] with strong anti-noise ability by increasing the size of convolution kernel in the first convolution layer, added Dropout [29] to the convolution layer, added Batch Normalization (BN) [30] layer by layer and other changes to VGG-net, and applied it to bearing fault diagnosis. Subsequently, they obtained a neural network named TICNN [31] with a stronger anti-noise ability by adjusting the convolution step size of the first convolution layer of WDCNN, and applied it to bearing fault diagnosis and 100% accuracy was obtained; however, when TICNN was used for frame structure fault diagnosis, the classification accuracy was reduced because the damage of adjacent structures has similar vibration data, the anti-noise ability was also reduced. Aiming at the problems existing in traditional feature extraction methods, based on convolutional neural network, Jing proposed a fault diagnosis method that directly learns features from the frequency data of vibration signals, and verified the effectiveness of the proposed method with the fault diagnosis of gearbox [32]. In order to identify the fault severity and fault direction of rolling bearings, Shao proposed a new integrated diagnosis method for intelligent fault diagnosis of rolling bearings based on deep learning theory [33]. In order to realize the automatic diagnosis of bearing faults, Hoang proposed a fault diagnosis method based on the deep structure of convolutional neural network. The method directly used vibration signals as input data, which is an automatic fault diagnosis system without any feature extraction technology and achieved very high accuracy and robustness in noise environment [34].

In terms of health monitoring, structural health monitoring based on deep learning has also been extensively studied. Based on the automatic recognition ability and strong classification ability of convolutional neural network, Abdeljaber et al., used 1DCNNs to conduct a fault diagnosis study on frame structure of the building model and obtained the expected diagnosis effect [35]; however, because there are few layers in the 1DCNN model and the size of convolution kernel is the same for each layer, the generalization ability of the network model is poor, the damage probability in the completely damage case does not reach 100%, and the damage probability of the no damage case reaches 9.85%. Yu et al. used improved bird swarm algorithm to optimize two-dimensional convolutional neural network to evaluate the torsional capacity of RC beams [36]. In order to overcome the influence of measurement noise and incomplete data on damage feature extraction, Guo et al. proposed a new method based on deep learning [37]. In order to solve the inaccuracy caused by data anomaly in structural health monitoring, Bao et al. proposed a data anomaly detection method based on computer vision and deep learning [38]. In order to overcome the shortcomings of the existing technology of surface crack diagnosis of concrete structure and improve the detection accuracy, Yu et al. proposed an automatic recognition method for surface condition identification of concrete structures based on vision by using deep learning theory [39]. To solve the problem of data anomaly in the data preprocessing stage of structural health monitoring, Tang et al. proposed a data anomaly detection method based on convolutional neural network [40].

When frame structures are used, they are often accompanied by various noises due to changes in the surrounding environment. Therefore, it is very important to accurately diagnose the faults of frame structures in the noise environment. In order to overcome the shortcomings of traditional fault diagnosis methods and improve the fault diagnosis accuracy of frame structure under strong noise conditions, in this paper, a convolutional neural network model named Improved Convolution Neural Networks with Training Interference (ITICNN) with strong anti-noise ability for frame structure fault diagnosis is proposed by reducing the input data length of TICNN, deepening the number of convolution layers, and adjusting a series of parameters of TICNN, and it was applied to the fault diagnosis of a 4-story building model [41] in the earthquake Laboratory of University of British Columbia. The research results show that the proposed method has high fault diagnosis accuracy and strong anti-noise ability, and the damage probability of frame structure is 0% in the case of no damage, and 100% in the case of completely damage.

In the rest of the paper, the structural frame model and its damage case are introduced in Section 2. Section 3 introduces the fault diagnosis process of the structural frame. Section 4 presents the improved TICNN and analyzes its performance. The fault diagnosis results of the frame structure are obtained in Section 5. Section 6 summarizes this paper and draws relevant conclusions.

## 2. Frame Structure

The model studied in this paper is a four-story steel frame structure was built by University of British Columbia [42]. Its 3D model is shown in Figure 1. It can be seen from Figure 1 that it is divided into four sides: south, east, north, and west, each side has the same structure distribution and the same location has the same code. Fifteen acceleration sensors were placed on the steel structure model. Starting from the first floor, three acceleration sensors were installed on each floor: one on the west side, one on the east side, and one near the center column. The acceleration sensors numbered No 1~No 3 were placed on the foundation of the bottom layer; the rests were placed on the top of each layer. Nine damage cases of frame structure can be simulated as shown in Table 1.

## 3. Fault Diagnosis Process of Frame Structure

Traditional fault diagnosis mostly uses neural network to process fault signals of the same frame structure to achieve multi classifications of faults. Due to the large amount of data and more classification results, when the signal similarity is high, this multi classification method often cannot achieve high accuracy, and requires high computer configuration. In contrast, using multiple binary classifications instead of multiple classifications requires less computer configuration, and has fast training speed and higher accuracy. Based on the superiority of binary classification, the fault diagnosis idea used in this article is takes the data of the frame structure in the two extreme cases of no damage (case 1) and the most serious damage (case 8) as training samples, then the improved ITICNN is used for binary classification, and the best model is sought by changing the number of convolution layers, the position of Dropout, and the size of convolution kernel during the training process. The specific fault diagnosis process is shown in Figure 2. As can be seen from Figure 2, the fault diagnosis process of frame structure is as follows:

(1)Fifteen acceleration sensors are placed on the four-story steel frame structure. The frame structure is operated according to the damage cases shown in Table 1. When the frame structure is intact (case 1), signals measured by acceleration sensors are marked as *U*, and signals measured under completely damage case (case 8) are marked as *D*. *U* and *D* can be expressed in the following form.
(1)$$\begin{array}{c} U=\left[U_{1},U_{2},\ldots \ldots \ldots .U_{j}\right]\\ D=\left[D_{1},D_{2},\ldots \ldots \ldots .D_{j}\right] \end{array}$$
where *j* is the number of acceleration sensor, *j* = (1~15).(2)Because the length of source data is relatively short, it is necessary to use data enhancement to increase the number of training samples so as to achieve the purpose that the length of input data is consistent with the number of samples in each training cycle. By using data enhancement, each source data are trimmed to 958 samples.
(2)$$958=\frac{K_{ij}-1024}{b}$$
where *K_ij_* is source signal length of the *j*th acceleration sensor under the *i*th damage case. The length of cropped fragment window is 1024, and *b* is window sliding step size. Since *K*_(*i*=1~5)*j*_ has 24,000 data points, *K*_(*i*=6)*j*_ has 60,000 data points, *K*_(*i*=7~9)*j*_ has 72,000 data points, so the step size in case 1 to case 5 is 24, the step size in case 7 to case 9 is 72 and the step size in case 6 is 60, so the data fragments are all 958.(3)The signal segments of *U_j_* (no damage) and *D_j_* (completely damage) between [−1,1] are normalized, and the normalized results can be written as
(3)$$\begin{array}{c} UNS=\left[UN_{1},UN_{2},\ldots \ldots \ldots UN_{j}\right]\\ DNS=\left[DN_{1},DN_{2},\ldots \ldots \ldots DN_{j}\right] \end{array}$$(4)After clipping and normalization, *UN_j_* and *DN_j_* are mixed to train the *j*th network. The training samples corresponding to each network are shown in Table 2.(5)All data fragments between [−1,1] are normalized under case 2~case 7 and case 9.(6)The saved *j*th-trained is used model to classify the data segment of the *j*th acceleration sensor under case *i*. The probability (denoted by *Pod_ij_*) of damaged segments to the entire signal segments is calculated, and the mean value of all *Pod_ij_* in the case *i* to is used to express the damage degree of case *i*. The larger the value, the more serious the damage. *Pod_ij_* can be calculated as follows.
(4)$$Pod_{ij}=\frac{k}{958}$$

## 4. ITICNN

TICNN is derived from WDCNN, and WDCNN is derived from VGG which is a visual neural network using 3 × 3 small convolution kernels layer by layer. VGG has the advantage of high accuracy, but the structure of two-dimensional convolution neural network directly used in vibration signal recognition and classification leads to low recognition rate due to the lack of receptive field. By increasing the length of the convolution kernel of the first convolution layer to 64 and the step size to 16, and using three small convolution layers to convolute the input signal of 2048 × 1, a one-dimensional convolution neural network named WDCNN with high recognition rate was obtained. On the basis of WDCNN, a network named TICNN can be obtained by changing the step size of the first large convolution layer to 8, and using the Dropout principle after the first large convolution layer to increase the network’s resistance to data loss, and at the same time increase the number of small convolution layers from 3 to 4. Zhang et al. proved that TICNN has strong anti-noise ability through comparative experiments. In order to achieve accurate fault diagnosis of frame structure under a strong noise interference environment, we chose TICNN as the neural network frame structure. However, when TICNN was used for frame structure fault diagnosis, the classification accuracy was reduced because the damage of adjacent structures has similar vibration data; in addition, the anti-noise ability was also reduced, so we improved TICNN.

In this paper, on the basis of TICNN, a network named ITICNN was obtained by adjusting the input of the first convolution layer of TICNN to 1024, changing the kernel size to 128 and step size to 2, and adding an intermediate transition layer with kernel size of 64, step size of 2, and channel of 32 after the first convolution layer. The intermediate transition layer was introduced to increase the size of convolution kernel to enhance the recognition ability of the network. Finally, the small convolution layer was increased to five layers to improve the anti-noise ability of the network. Since the ability of Dropout to suppress fitting in the convolutional layer is not obvious, Dropout was added to the full connection layer to prevent over-fitting. The structure of ITICNN is shown in Figure 3.

In Figure 3, P represents Max-pooling, C represents the length, T represents the number of channels, G represents the good type, and B represents the bad type. After each pooling layer in ITICNN, the BN operation was used; the BN layer can standardize the data to enhance the recognition ability of the network.

### 4.1. Convolution Operation

The convolution process of one-dimensional convolutional neural network does not need to move to the right. It only needs to traverse the entire input sequentially according to a certain step size. The receptive field of the first convolution layer is the length of the first convolution layer’s kernel. The convolution operation is shown in Figure 4.

In Figure 4, we suppose the length of input layer is *N*, the size of the convolution kernel is *M*, and the size of moving step is 1, then the length after convolution is *N* − *M* + 1. In the convolution layer, the one-dimensional forward propagation can be defined by Equation (5) as follows
(5)$$x_{k}^{l}=b_{k}^{l}+\overset{N_{l-1}}{\underset{i=1}{\sum }}(W_{ik}^{l-1}\times S_{i}^{l-1})$$
where $x_{k}^{l}$ is the input of the *k*th neuron in the *l*th layer;$\;b_{k}^{l}$ is the deviation of the *k*th neuron in the *l*th layer;$\;S_{i}^{l-1}$ is the output of the *i*th neuron in the *l −* 1th layer;$\;W_{ik}^{l-1}$ is the weight of the *i*th neuron in the *l* − 1 layer.

### 4.2. Batch Normalization

BN is also called batch normalization processing. BN operations are added after each convolutional layer in ITICNN until the end of the last convolutional layer. Its principle is shown in Figure 5.

In batch normalization process, there are three hidden layers between input and output. As shown in Figure 5, *X* is the input of this layer and *Y* is the output of this layer. The network has three hidden layers, which are composed of h_1_ and h_2_, h_2_ and h_3_, and h_3_ and h_4_, respectively. The relationship between the output of first hidden layer and the input of h_1_ can be expressed as the following equations:(6)$$\mu_{\beta }=\frac{\sum_{i=1}^{n}\left(W_{i}X_{i}+b\right)}{n}=\frac{\sum_{i=1}^{n}W_{i}X_{i}}{n}+b$$
(7)$$\sigma_{\beta }^{2}=\frac{\sum_{i=1}^{n}{\left(W_{i}X_{i}+b-\mu_{\beta }\right)}^{2}}{n}$$
(8)$$S_{1i}-\mu_{\beta }=W_{i}X_{i}-\frac{\sum_{i=1}^{n}W_{i}X_{i}}{n}$$
(9)$$S_{2i}=\frac{S_{1i}-\mu_{\beta }}{\sqrt{\sigma_{\beta }^{2}+\int }}=\frac{W_{i}X_{i}-\frac{\sum_{i=1}^{n}W_{i}X_{i}}{n}}{\sqrt{\sigma_{\beta }^{2}+\int }}$$
(10)$$S_{3i}=\gamma_{i}S_{2i}+\beta$$
where *n* is the length of mini-batch in the BN; *X_i_* is the *i*th input; *S*_1*i*_, *S*_2*i*_, and *S*_3*i*_ are the *i*th output of the 1st, 2nd, and 3rd hidden layers; *W_i_* is the *i*th weight of BN operation in this layer; *b* is the offset; $\mu_{\beta }$ and $\sigma_{\beta }^{2}$ are the expected and variance of a small batch input, respectively; ϵ is a decimal that tends to 0, but it cannot be set to 0 because when it is equal to 0, Equation (9) does not hold; *γ* and *β* are the parameters to be learned.

### 4.3. Dropout

The main idea of Dropout is to randomly lose every neuron that propagates forward with a probability *P* when training the neural network, this probability *P* obeys the Bernoulli distribution. The model obtained in this way has stronger generalization ability, because it is not restricted by a certain local feature. Its principle is shown in Figure 6.

The network calculation formula when Dropout is not used can be expressed as:(11)$$Z_{i}^{l+1}=W_{i}^{l+1}y^{l}+b_{i}^{l+1}$$
(12)$$y^{l+1}=f\left(Z_{i}^{l+1}\right)$$

The network calculation formula when Dropout is used can be expressed as:(13)$$r^{l}\sim Bernoullia\left(P\right)$$
(14)$$Y^{l}=r^{l}\times y^{l}$$
(15)$$Z_{i}^{l+1}=W_{i}^{l+1}Y^{l}+b_{i}^{l+1}$$
(16)$$y^{l+1}=f\left(Z_{i}^{l+1}\right)$$
where $y^{l}$ represents the output of the *i*th neuron in the *l*th layer; $Z_{i}^{l+1}$ is the output of the *i*th neuron in the *l* + 1th layer; $b_{i}^{l+1}$ represents the bias after propagation of the *i*th neuron in the *l* + 1th layer; $W_{i}^{l+1}$ represents the weight of the *i*th neuron in the *l* + 1th layer; $r^{l}$ obeys Bernoulli distribution, $y^{l+1}$ is the input of the next layer after $Z^{l+1}$ is activated; and *f* is the activation function.

### 4.4. Parameters Design

According to the network structure shown in Figure 3, the parameters of each layer of network were designed as shown in Table 3. In addition, in network training BN was used in each layer after pooling layer, and Dropout operation was used in the full connection layer, and its parameter is 0.5.

### 4.5. Model Training Results

The training of convolutional neural network proposed in this paper was carried out on a laptop equipped with Intel Corei7-4710MQ CPU, NVIDIA GT940M 2G GPU, and 12G memory. In order to show the training results of ITICNN, the data of the 13th acceleration sensor were randomly selected from 15 types of acceleration sensors as samples for training. During training, the data processing and convolutional neural network operating environments were Ten-sorflow 2.4.8, keras 3.4.2, and Python 3.8.0; the batch size was 128 and the training epoch was 150. In addition, this paper adopted Adam to optimize the parameters of the neural network. Because Adam has the adaptive ability, and the use of Adam’s default initial learning rate for adaptive learning has the advantages of fast training speed, obvious changes in the training curve, and strong anti-over-fitting ability, so this paper used the default initial learning rate of the operating environment. The accuracy and loss shown in Figure 7 were obtained through training.

It can be seen in Figure 7 that the accuracy of training and testing increases rapidly when the training epoch reaches 25; it is close to the stable value when the training epoch reaches 40, and it is stable at 100%. At the same time, the objective function value (loss) also drops very fast, when the training epoch reaches about 30 the objective function value tends to be stable. When the training epoch reaches 80 the objective function curves of training and testing coincide, when the training epoch reaches 100 the objective function value drops to 10^−5^. The following conclusions can be drawn from Figure 7:(1)There is no over-fitting phenomenon during training;(2)High training accuracy and small objective function value can be obtained;(3)The training speed is fast, and the model can be saved when training epoch reaches about 80.

### 4.6. Confusion Matrix Analysis

The confusion matrix is also called error matrix. The parameters in the confusion matrix are often used in training of convolutional neural network to calculate Precision, Accuracy, Recall, and F1-Score of the network. When using confusion matrix to calculate network indicators, it is often required that the number of samples in each situation in the data set is close so the results obtained are more reliable. In this paper, the convolution neural network matched with the 13th acceleration sensor selected randomly from the 15 accelerometers used to obtain the confusion matrix. The confusion matrix is shown in Figure 8.

The Precision, Accuracy, and Recall of the network model can be obtained from Figure 8. The formula for calculating the Accuracy is as follows:(17)$$\mathrm{Accuracy}=\frac{\mathrm{TP}+\mathrm{TN}}{\mathrm{TP}+\mathrm{FT}+\mathrm{TN}+\mathrm{FN}}$$
where True Positive (TP) is a positive sample predicted by the model as positive; True Negative (TN) is a negative sample predicted by the model as negative; False Positive (FP) is a negative sample predicted by the model as Positive; False Negative (FN) is a positive sample predicted by the model as negative. It can be seen from Figure 8 that TP = 1, FP = 0, TN = 1, and FN = 0.

The calculation formulas of Precision and Recall are as follows:(18)$$\mathrm{Precision}=\frac{\mathrm{TP}}{\mathrm{TP}+\mathrm{FP}}$$
(19)$$\mathrm{Recall}=\frac{\mathrm{TP}}{\mathrm{TP}+\mathrm{FN}}$$

Through the above formulas, we can know that Accuracy = 100%, Precision = 100%, and Recall = 100%, which can draw a conclusion that the diagnosis result is more reliable. In addition, the average evaluation index F1-Score of Precision and Recall can be calculated by the following formula:(20)$$F1\text{-Score}=2\times \frac{\mathrm{Precision}\times \mathrm{Recall}}{\mathrm{Precision}+\mathrm{Recall}}$$

From Formula (20), we can obtain F1-Score = 100%, which shows that the effect of using ITICNN to train fault data of two classifications is optimal. At the same time, it can be seen from Figure 8 that Accuracy is higher when using the two-class method for fault diagnosis.

### 4.7. Comparison of Anti-Noise Ability

In this section, the anti-noise performance of ITICNN was studied, and the experimental results were compared with 1DCNN, WDCNN, and TICNN. In the experiment, the vibration signals measured by the 14th acceleration sensor in case 1 and case 8 were used. In the contrast experiments, the most intuitive method was used to study the anti-noise ability of the four network models. Firstly, the data collected by the 14th acceleration sensor in case 1 and case 8 were trained and the trained model was saved, then the data with different noises and known labels were tested by using the saved model, finally Accuracy was calculated and compared with the value calculated by the confusion matrix. The signal-to-noise ratio (SNR) is defined as follows:(21)$${\mathrm{SNR}}_{}=\frac{Signpower}{noisepower}$$
where SNR is signal-to-noise ratio, *Signpower* is signal power, and *noisepower* is noise power. The following equation can be obtained by changing the unit of SNR into dB:(22)$${\mathrm{SNR}}_{\mathrm{dB}}=10\log\mathrm{SNR}$$

The comparison results are shown in Figure 9. It can be seen from Figure 9 that ITICNN’s anti-noise ability is about 3~4% stronger than TICNN, the anti-noise ability of TICNN is 28~30% greater than WDCNN, and the anti-noise ability of ITICNN is 30~32% stronger than 1DCNN. At the same time, it can be clearly seen from the figure that when the Dropout layer is not used the anti-noise ability of the model proposed in this paper is reduced by 20~25%, when the BN layer is not used the accuracy of the model proposed in this paper is reduced by 30–35%, which basically coincides with the result of 1DCNN. It can be concluded that BN layer is the key to improve the anti-noise ability of ITICNN. At the same time, adding a Dropout layer between the first large convolutional layer and the second convolutional layer can improve anti-noise ability of ITICNN. Therefore, ITICNN is superior to TICNN, WDCNN, and 1DCNN in terms of anti-noise performance.

## 5. Fault Diagnosis Results

In order to verify the effectiveness of the ITICNN model proposed in this paper, an experimental study was carried out. In the experiment, the 12 saved ITICNN training models were used to classify the data segments of the corresponding acceleration sensors in each damage situation, then the value of each *Pod_ij_* was calculated according to the number of the damaged segments and total number segments in the predicted fragments. Finally, the value of each *Pod_ij_* and their average value of the 12 acceleration sensors under 9 kinds of damage cases were obtained. The results are shown in Table 4. In Table 4, F is floor, W is west, C is center, E is east.

It can be clearly seen from Table 4 that the vibration signals collected by the acceleration sensors with ID of 4, 5, 6, 8, 9, and 13 have little change in case 1~case 6, and in other damage cases there is a problem that different damage cases have the same value of damage probability (*Pod_ij_*). Therefore, it is not feasible to use different *Pod_ij_* of a single acceleration sensor to judge the damage condition of frame structure. In addition, from the change in the average value of *Pod_ij_* in Table 4, it can be seen that with the increase in the damage degree of the frame structure, the average value of *Pod_ij_* is also increasing, and there are obvious differences among the values. Therefore, the damage degree of the frame structure can be clearly judged according to the average value of *Pod_ij_*.

In order to verify the advantages of the network model proposed in this paper, under the same conditions the 1DCNN proposed by Abdeljaber et al. in 2018 was used to conduct experimental comparative research on the frame structure, and histogram of the comparison results between ITICNN and 1DCNN was obtained, as is shown in Figure 10.

It can be seen from the comparison results in Figure 10 that when the damage of frame structure under case 1 (no damage) and case 8 (completely damage), the diagnosis results of ITICNN model are 0% and 100%, respectively, and the results have no error. However, when using 1DCNN, there are certain diagnosis errors in both cases. Therefore, the fault diagnosis ability of ITICNN proposed in this paper is better than that of 1DCNN. In addition, it can be seen from Figure 10 that 1DCNN can distinguish damage case 1~case 7, but it is difficult to distinguish damage case 8 and case 9 (the results in two cases are too close), this is due to the poor noise resistance and generalization ability of 1DCNN. When using ITICNN for fault diagnosis of frame structure, not only can case 1~case 7 be distinguished obviously, but also the difference value of *Pod_ij_* between case 8 and case 9 is 5.66%, so it is easy to distinguish case 8 and case 9. This fully shows that ITICNN is superior to 1DCNN in noise resistance and generalization ability, and better results can be obtained by using ITICNN for fault diagnose of structural frame.

## 6. Conclusions

In order to realize the accurate fault diagnosis of structure frame under strong noise condition, a convolutional neural network model named ITICNN with strong anti-noise ability is obtained by improving TICNN. Based on the proposed ITICNN, aiming at the fault diagnosis problem of the 4-story steel structure frame model, the model is trained by using data sets collected under two extreme damage conditions (no damage and completely damage), and the damage probability under other damage conditions is predicted by using the trained model, and the ideal results are obtained. Accurate fault diagnosis of 4-story steel structure frame model can be realized by comparing the average damage probability of various damage cases obtained by the fault diagnosis method proposed in this paper with the actual damage situations of frame structure. In order to verify the advantages of ITICNN, experimental studies are compared with TICNN, WDCNN, and 1DCNN under the same conditions. The results show that ITICNN is superior to TICNN, WDCNN, and 1DCNN in terms of anti-noise performance, and there are obvious differences in damage probability under different damage conditions. It can easily distinguish all damage cases, and the fault diagnosis performance is obviously better than that of 1DCNN. Therefore, the ITICNN proposed in this paper can be used for fault diagnosis of frame structure effectively, and the diagnosis results are more accurate.

## Figures and Tables

**Figure 1 sensors-22-09427-f001:**
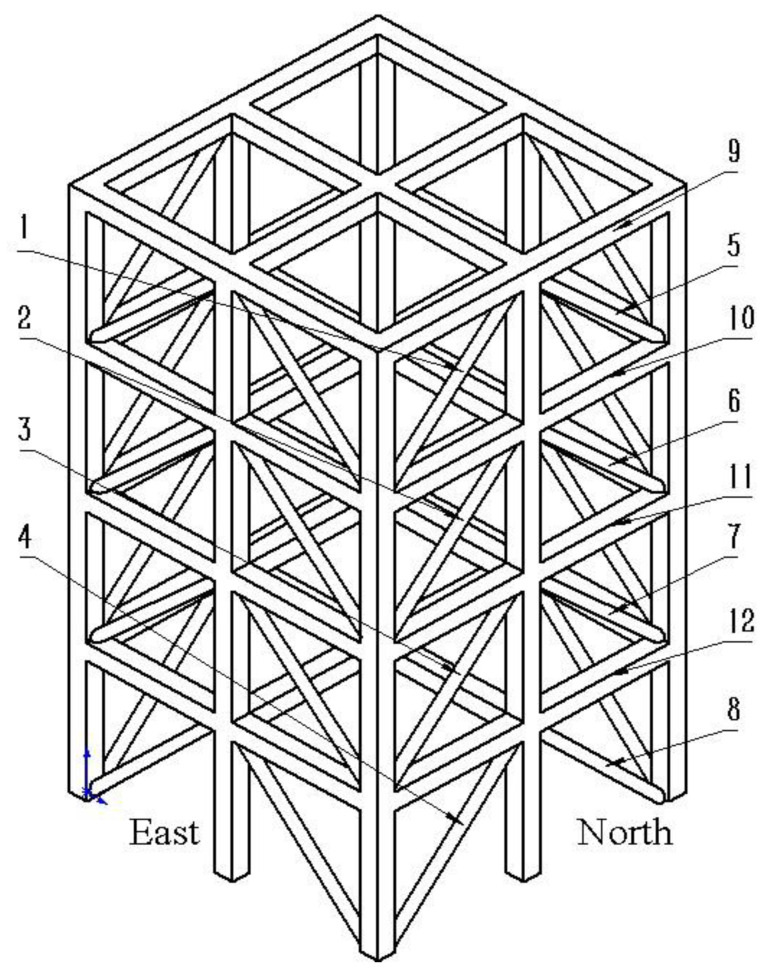
The 3D model of the four-story steel frame structure.

**Figure 2 sensors-22-09427-f002:**
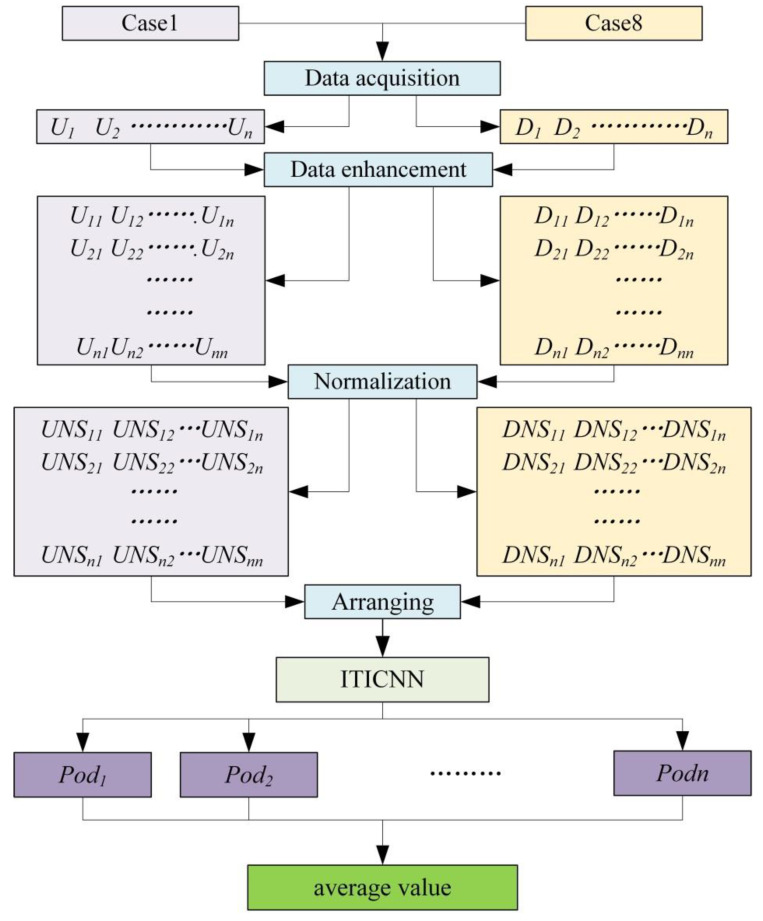
Fault diagnosis process.

**Figure 3 sensors-22-09427-f003:**
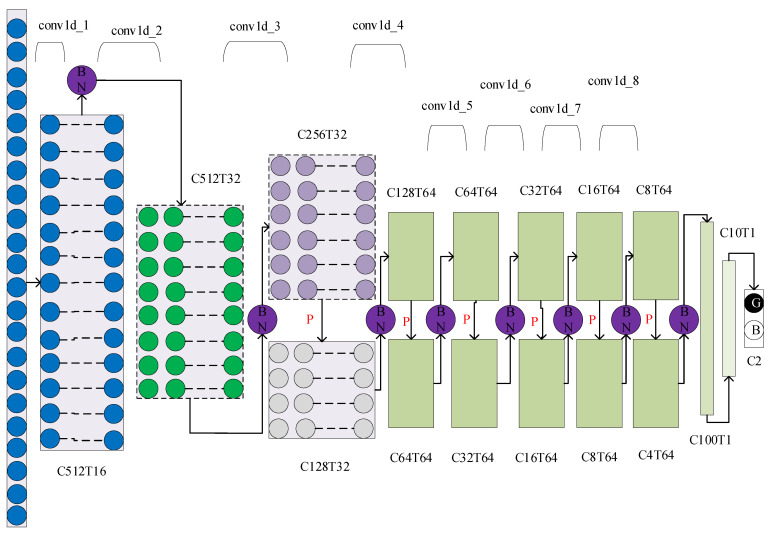
The structure of ITICNN.

**Figure 4 sensors-22-09427-f004:**
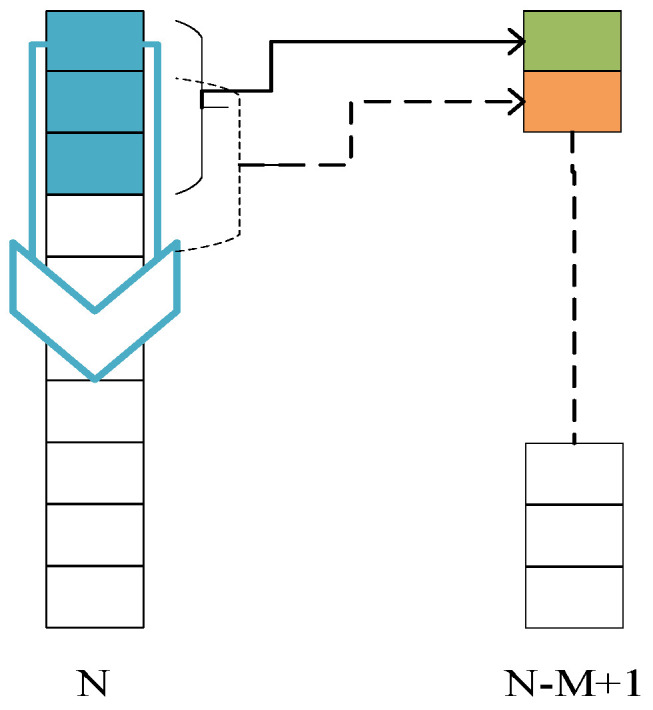
Convolution operation.

**Figure 5 sensors-22-09427-f005:**
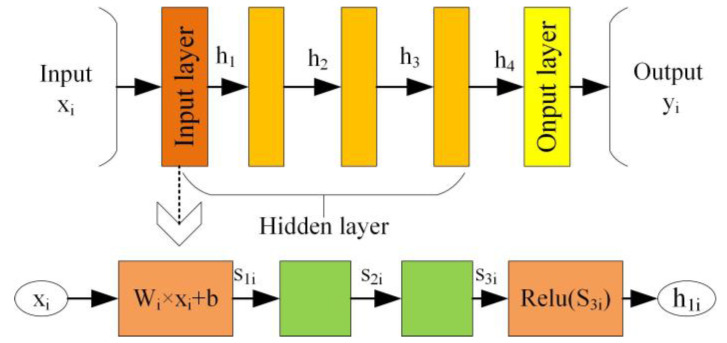
Schematic diagram of BN.

**Figure 6 sensors-22-09427-f006:**
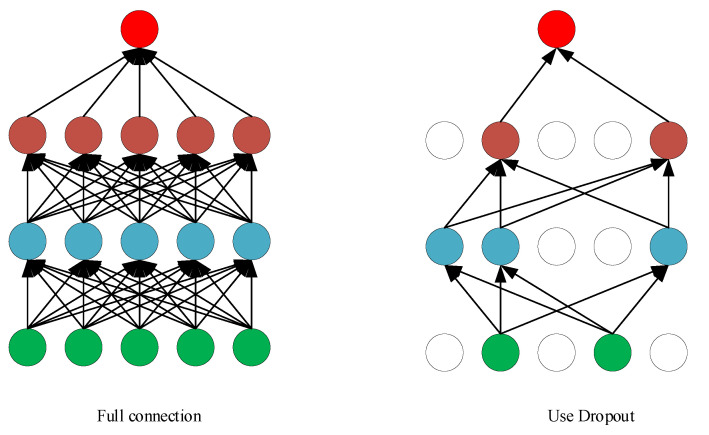
The principle of Dropout.

**Figure 7 sensors-22-09427-f007:**
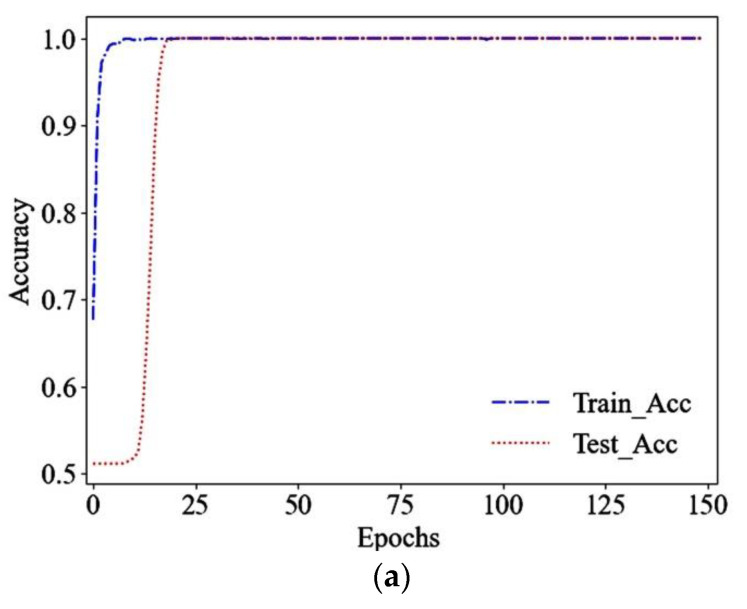
Accuracy and loss. (**a**) Accuracy; (**b**) loss.

**Figure 8 sensors-22-09427-f008:**
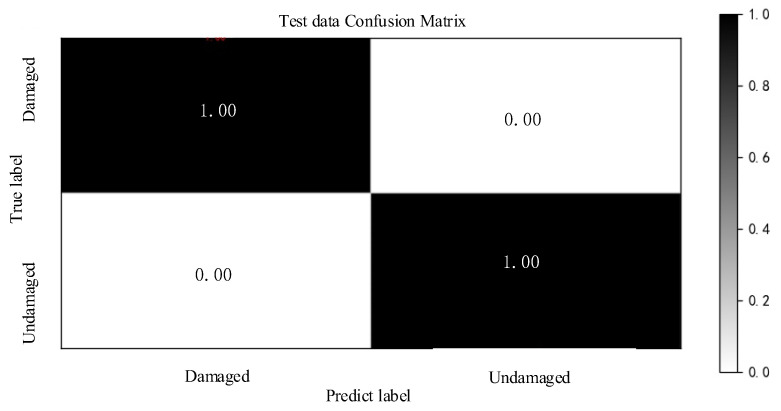
Confusion matrix.

**Figure 9 sensors-22-09427-f009:**
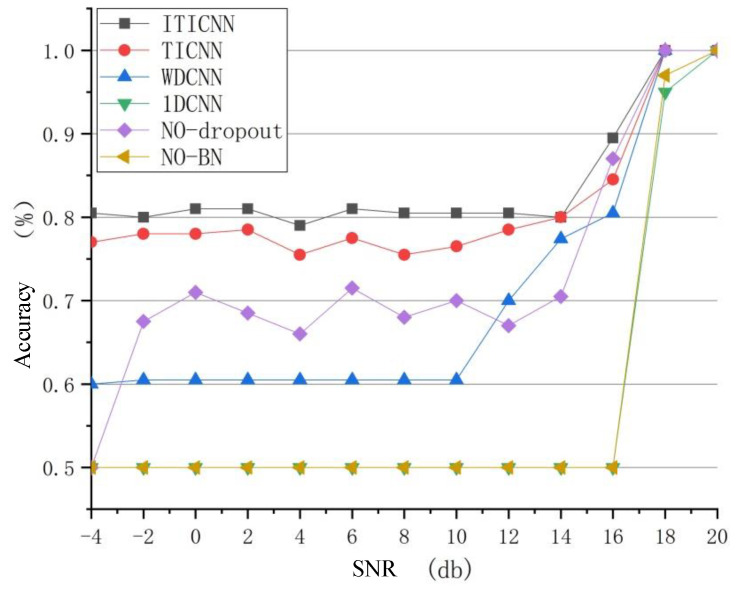
Comparison results of anti-noise ability.

**Figure 10 sensors-22-09427-f010:**
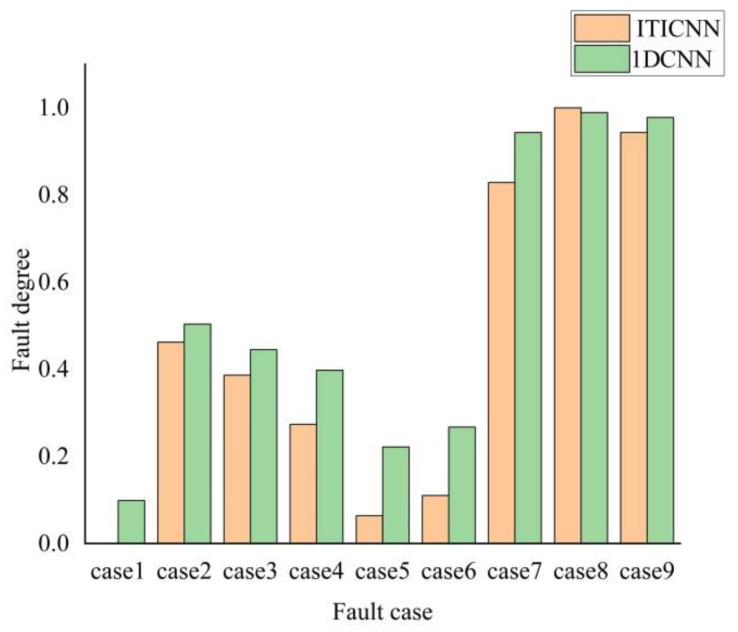
Comparison results between ITICNN and 1DCNN.

**Table 1 sensors-22-09427-t001:** Damage cases of frame structure.

Damage Cases	Specific Operation
1	No damage
2	Remove structures numbered 1–8 in the east
3	Remove structures numbered 1–4 in the east
4	Remove structures numbered 1 and 4 in the east
5	Remove structure numbered 1 in the east
6	Remove structures numbered 3 and 7 in the north
7	Remove structures numbered 1–8 in all 4 sides
8	Loosen structures numbered 9, 10, 11 and 12 in the east on the basis of case 7
9	Loosen structures numbered 11 and 12 in the east on the basis of case 7

**Table 2 sensors-22-09427-t002:** Training samples.

Neural Network Number (*j*)	Acceleration Sensor Number	Total Sample Size	Training Set	Test Set
1	4	1916	1341	575
2	5	1916	1341	575
3	6	1916	1341	575
4	7	1916	1341	575
5	8	1916	1341	575
6	9	1916	1341	575
7	10	1916	1341	575
8	11	1916	1341	575
9	12	1916	1341	575
10	13	1916	1341	575
11	14	1916	1341	575
12	15	1916	1341	575

**Table 3 sensors-22-09427-t003:** Parameters of ITICNN.

Layer Type	Channels	Kernel/Step	Output Size	Zero Padding
Convolutional layer 1	16	128/2	512 × 16	yes
Convolutional layer 2	32	64/2	256 × 32	yes
Convolutional layer 3	32	3/1	256 × 32	yes
Max-pooling layer	32	2/2	128 × 32	no
Convolutional layer 4	64	3/1	128 × 64	yes
Max-pooling layer	64	2/2	64 × 64	no
Convolutional layer 5	64	3/1	64 × 64	yes
Max-pooling layer	64	2/2	32 × 64	no
Convolutional layer 6	64	3/1	32 × 64	yes
Max-pooling layer	64	2/2	16 × 64	no
Convolutional layer 7	64	3/1	16 × 64	yes
Max-pooling layer	64	2/2	8 × 64	no
Convolutional layer 8	64	3/1	8 × 64	yes
Max-pooling layer	64	2/2	4 × 64	no
Full connection layer	1	100	100	no
Fully connection layer	1	10	10	no
Softmax	1	2	2	no

**Table 4 sensors-22-09427-t004:** Fault diagnosis results.

Sensor ID	Sensor Location	1	2	3	4	5	6	7	8	9
4	1st F W	0%	0%	0%	0%	0%	2.71%	2.73%	100%	33.51%
5	1st F C	0%	0%	0%	0%	0%	0%	0.84%	100%	100%
6	1st F E	0%	0%	0%	0%	0%	0%	100%	100%	100%
7	2st F W	0%	77.77%	36.95%	16.08%	1.04%	14.2%	98.75%	100%	98.75%
8	2st F C	0%	0%	0%	0%	0%	0%	100%	100%	100%
9	2st F E	0%	0%	0%	0%	0%	0%	98.96%	100%	100%
10	3st F W	0%	100%	97.7%	95.2%	66.28%	100%	100%	100%	100%
11	3st F C	0%	99.9%	39.18%	9.4%	0%	14.73%	98.12%	100%	99.79%
12	3st F E	0%	100%	100%	27.17%	0%	0.1%	100%	100%	100%
13	4st F W	0%	0%	0%	0%	0%	0%	95.19%	100%	100%
14	4st F C	0%	77.04%	89.77%	80.17%	0%	0%	100%	100%	100%
15	4st F E	0%	100%	100%	100%	9.09%	0%	100%	100%	100%
Average		0%	46.23%	38.63%	27.34%	6.37%	11.00%	82.88%	100%	94.34%

## Data Availability

Data sharing not applicable.

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
