# Peer review of "Frame Structure Fault Diagnosis Based on a High-Precision Convolution Neural Network"

_sensors, 2022, doi:10.3390/s22239427_

Round 1
Reviewer 1 Report
Paper is interesting, well written and is of high technical contents explained and validated well
Author Response
Thank you very much for your comment.hank you for your comment.
Reviewer 2 Report
In this paper, the authors proposed an improved TICNN model for frame structure fault diagnosis. The experimental results are performed to verify the performance of the proposed method. However, there are still some issues that need to be addressed.
1. There should be one paragraph in the last of Introduction to summary the main part of this paper.
2. In the introduction part, the authors should add deep learning algorithms for rotating machinery intelligent diagnosis or some review papers and benchmark studies.
3. The training details should be stated clearly, for example, learning rate, batch size, etc.
4. There should contain other metrics to verify the performance of the proposed for such a binary classification problem. For example, F1-score.
5. “SNR” and “SNR” should be unified.
6. “bd” and “dB” should be unified.
Author Response
Thank you very much for your comments. We upload our reply in the form of an attachment.

Reviewer 3 Report
I reviewed the paper entitled “Frame structure fault diagnosis based on a high 3 precision convolution neural network”. I have the following suggestion for improvement of the paper.
· The literature review is quite scant, and it requires a thorough explanation, why the network is selected, and what motivated authors for framing this structure. A good example of depicting the research direction is showcased in following papers, I advise authors to read the flow of the mentioned papers.
1. Sharma, Ankit Kumar, et al. "Voltage stability assessment using artificial neural network." 2018 IEEMA Engineer Infinite Conference (eTechNxT). IEEE, 2018.
2. Li, C., Xiong, G., Fu, X., Mohamed, A. W., Yuan, X., Al-Betar, M. A., & Suganthan, P. N. (2022). Takagi–Sugeno fuzzy based power system fault section diagnosis models via genetic learning adaptive GSK algorithm. Knowledge-Based Systems, 255, 109773.
· Fault diagnosis process depicted in figure 2 is quite complex. Hence, I request the authors that this should be explained in a lucid manner.
· Also in figure 7, please split the graph into two folds, i.e., separate it for loss and accuracy. Also, in figure 10 you have to mention the axis. Also, I feel the representation of the results as well as the methodological explanation of the development of the network requires description.
Author Response

(The authors gave the same response as above.)

Round 2
Reviewer 2 Report
I do not have any further comments.
Reviewer 3 Report
Comments are addressed.